# Assessing the completeness and accuracy of South African National Laboratory CD4 and viral load data: a cross-sectional study

Ingrid Valerie Bassett,[1,2,3,4,5] Mingshu Huang,[3,4,6] Christie Cloete,[7] Sue Candy,[8] Janet Giddy,[7] Simone Claire Frank,[2,3] Robert A Parker[3,4,5,6]

These data were presented in part at the 21st International AIDS Conference (AIDS 2016) 18 July to 22 July 2016 in Durban, South Africa.

For numbered affiliations see end of article.

**Correspondence to**
Dr Ingrid Valerie Bassett;
ibassett@partners.org

## ABSTRACT

**Objective** To assess the accuracy of the South African National Health Laboratory Services (NHLS) corporate data warehouse (CDW) using a novel data cross-matching method.

**Methods** Adults (≥18 years) on antiretroviral therapy (ART) who visited a hospital-based HIV clinic in Durban from March to June 2012 were included. We matched patient identifiers, CD4 and viral load (VL) records from the HIV clinic's electronic record with the NHLS CDW according to a set of matching criteria for patient identifiers, test values and test dates. We calculated the matching rates for patient identifiers, CD4 and VL records, and an overall matching rate.

**Results** NHLS returned records for 3498 (89.6%) of the 3906 individuals requested. Using our computer algorithm, we confidently matched 3278 patients (83.9% of the total request). Considering less than confident matches as well, and then manually reviewing questionable matches using only patient identifiers, only nine (0.3% of records returned by NHLS) of the suggested matches were judged incorrect.

**Conclusions** We developed a data cross-matching method to evaluate national laboratory data and were able to match almost 9 of 10 patients with data we expected to find in the NHLS CDW. We found few questionable matches, suggesting that manual review of records returned was not essential. As the number of patients initiating ART in South Africa grows, maintaining a comprehensive and accurate national data repository is of critical importance, since it may serve as a valuable tool to evaluate the effectiveness of the country's HIV care system. This study helps validate the use of NHLS CDW data in future research on South Africa's HIV care system and may inform analyses in similar settings with national laboratory systems.

## INTRODUCTION

South Africa has the largest HIV treatment programme in the world, with >3.1 million people on antiretroviral therapy (ART).[1] The government has expanded its national programme in recent years in a transition to 'country ownership' from the previous non-governmental organisations and private

### Strengths and limitations of this study

► This is the first analysis to propose a novel method for examining the completeness and accuracy of records related to HIV care from a national data source.

► We developed a comprehensive and self-contained algorithm using commonly available patient identifiers (first name, surname, date of birth, gender) that may inform future analyses focusing on linkage to and retention in HIV care, and this methodology may also apply to data matching analyses in similar settings, as many sub-Saharan African countries have some sort of national laboratory system.

► National Health Laboratory Service requirements for submitting identifiers with laboratory requisitions during the study period were not strict enough to allow uniformly perfect matching; thus, we had to create extensive matching categories to cover the range of match types and quality.

► While we considered our patient identifier, CD4 and viral load test record matching criteria detailed and comprehensive, a different team might develop an alternative set of rules and designations, and classify specific results differently.

► We had a large range of patient identifier matching criteria for what we considered an adequate match, while these criteria were discussed at length, they ultimately were subjective decisions.

clinics.[2–4] As HIV care transitions to the public sector and the number of patients initiating ART grows, maintaining comprehensive and accurate patient data is of critical importance. Reliable and valid national data become increasingly useful for evaluating linkage to and retention in HIV care, for monitoring patients longitudinally across clinic sites, and for assessing the quality of care at the national level.

Patients undergoing HIV treatment at public and semiprivate health centres in South Africa have routine blood samples

sent to a National Health Laboratory Service (NHLS) laboratory for testing; these data are then stored at a central repository in the NHLS corporate data warehouse (CDW). NHLS data have previously been used to evaluate the effectiveness of certain government-funded HIV programmes,[5–7] identify patterns of the TB epidemic[8 9] and determine cancer incidence rates among HIV-infected individuals.[10] CD4 count and viral load (VL) records serve as indicators of being in HIV care, as these are monitored regularly while patients are receiving ART. However, NHLS CDW data have not been assessed to determine utility specifically for identifying and tracking patients in HIV care. While previous studies have compared mortality records between South African civil registration and clinics to evaluate the completeness of national mortality data,[11 12] no such comparison has been performed between CD4 and VL records for patients in HIV care.

We assessed the completeness and accuracy of the NHLS CDW for tracking patients using a cohort of patients who visited McCord Hospital's HIV clinic during a 3-month period just prior to clinic closure due to loss of funding. We present here a method developed to match patients based on McCord Hospital patients' identifiers, CD4 records and VL values prior to transfer to data provided to us by the NHLS.

## METHODS
### Study site
McCord Hospital was a semiprivate, general hospital in KwaZulu-Natal serving a predominantly urban population from the greater Durban area. The Sinikithemba HIV clinic at McCord, which became a President's Emergency Plan for AIDS Relief (PEPFAR)-funded site in 2004, was an integral part of the South African ART scale-up and initiated over 10 000 patients on ART.[13] Sinikithemba served a predominantly African, Zulu-speaking population. The clinic had a monitoring and evaluation team and an electronic medical record. Due to loss of PEPFAR funding, the clinic closed in 2012.

All patients who returned to the clinic for clinical appointments, laboratory tests or pharmacy refills from 12 March to 30 June 2012 were referred for transfer to clinics in the Durban area. Data collected at the time of transfer included name, gender, date of birth (DOB), most recent pretransfer CD4 count and VL values and dates. We have previously reported on the Sinikithemba transfer process evaluating linkage to initial transfer clinic visit and patient attitudes about their transfer experience using telephone surveys and clinic visits.[13 14]

### Study population
We studied adults' ≥18 years on ART who visited the HIV clinic during the transfer period. Routinely collected programmatic data were used.

## National Health Laboratory Service
The NHLS was established in 2001 and supports national and provincial health departments in South Africa. It is the largest diagnostic pathology service in the country, providing laboratory and related services to over 80% of the population through a national network of laboratories.[5] The NHLS performs all public sector CD4 and VL monitoring and maintains a CDW that serves as a national repository for laboratory data from the public sector. Healthcare workers at public health facilities complete laboratory requisition forms which accompany each sample submitted to the CDW. All data, including patient identifiers, name of facility, date of sample and tests requested, are sent to the CDW and are captured electronically by the NHLS information system in real time. The CDW has developed an algorithm which uses both rules and probabilistic matching based on demographic attributes using fuzzy logic.[15 16] This is applied to all test data at time of entry and results in a Master Patient Index within the CDW.

## Data collection and processing
We sent a list of all 4257 McCord Hospital transfer patients with corresponding patient identifiers (first name, surname, DOB, gender) to the NHLS for matching of laboratory records (online supplementary figure 1A,B). We also included an internal study ID to identify each patient so that the NHLS could determine which records they were providing matched our requested records. The NHLS extracted data in October 2014. McCord Hospital data were matched against the entire CD4 and VL datasets for KwaZulu-Natal province from 1 November 2010 to 31 October 2014. To minimise the data lost when exchanging between systems, the NHLS has checks in place to ensure that the number of records sent by the Laboratory Information System (LIS) interface are processed into the CDW. In the event of system failures, there is the ability to re-submit data from the LIS. Trend reporting of test volumes over time also assists with data gaps. To assist with the matching process, we also sent last known CD4 and VL values and dates recorded in the electronic medical record at McCord Hospital. We received two datasets (CD4 count and VL) containing potential matches from the NHLS. These datasets had 16 340 and 18 677 records from 3774 patients. We performed three separate matching analyses using patient identifiers (first name, surname, DOB, gender), CD4 counts and test dates, and VL values and test dates. In each analysis, we assessed the quality of the match within our internal study ID for each patient; thus, we assessed how well the data provided by the NHLS using their probabilistic matching technique represented a true match. From the original 4257 patient list, duplicated study IDs (n=12) and patients <18 years on 30 June 2012 (n=337) were removed prior to matching. Two patients who had neither a CD4 count nor VL record from McCord Hospital were also removed. This left a cohort of 3906 patients to match based on patient identifiers. For the CD4 matching

analysis, we removed 1 patient who did not have CD4 data in the McCord database, for a cohort of 3905 patients. We removed 297 patients who did not have VL data in the McCord database (missing VL data may reflect a test not being performed or patients recently initiated on ART who had not yet met guidelines for undergoing a VL test), resulting in a cohort of 3609 patients for the VL record matching analysis.

### Matching of records between NHLS and McCord datasets

We performed our matching analysis in three stages; first, we cross-checked patient identifiers between the McCord and NHLS datasets to determine the distribution of optimal identifier matching, using all records for a particular individual prior to clinic closure. Next, we assessed the reported CD4 and VL records separately, independent of patient identifiers. Lastly, we considered the best test record match from a particular internal study ID number in conjunction with the patient identifier match for that specific record to determine the overall distribution of matching based on both test records and patient identifiers. In this final matching analysis, the patient identifier match was determined for the better match on either CD4 or VL. If the test match quality was the same, we used the better patient match of the two test records.

#### Matching using patient identifiers

Within each internal study ID for each patient, we used surname, first name, DOB and gender to assess the quality of the match between the NHLS CDW and the McCord data record. Based on a detailed set of matching criteria (online supplementary table 1), we classified patient study IDs into five general matching categories: confident, likely, likely despite keying errors, possible and other. If corresponding patient identifiers fell into the latter two categories, they were reviewed manually; otherwise they were considered an adequate match and not reviewed. The manual review processes consisted of an independent review by two authors (IVB, SCF), with a third 'tiebreaker' review by another author (RAP) for any discordant matching designations.

#### Matching based on test results

We had a cohort of 3905 patients for the CD4 record matching analysis and 3609 patients for the VL matching analysis. If the CD4 count in the McCord record and a corresponding NHLS CDW record were an exact match, we compared the McCord test data to the two dates provided by the NHLS (test date and record date) for consistency (online supplementary table 2). When the dates were consistent (exact match, month and day reversed, dates differed by less than 7 days, dates differed by one of year, month or day), we considered the records a confident match. If the CD4 counts from corresponding McCord and NHLS CDW records differed, but there was an exact match on dates, we considered the records a possible match. If the dates were not consistent, we considered the records an unlikely match, even if the

CD4 values matched. Records containing both discrepant CD4 values and mismatching dates were considered no match. Following these same criteria, we categorised corresponding NHLS CDW and McCord VL records as confident, possible or unlikely matches. Because VL is often reported as undetectable, we had to use a somewhat looser criterion for considering the VL result an exact match (online supplementary table 2).

### Matching based on patient identifier, conditional on matching based on a test result

After matching CD4 and VL values and dates, we assessed the accuracy of the patient identifier information based on the specific record used for the test matching. When there were equally good matches for both the CD4 and VL test, we used the better of the two patient matches for this classification.

### Patient and public involvement

Neither patients nor the public were involved in developing this project.

## RESULTS

### Cohort characteristics

Of 3906 participants included in the analysis, 41% of the cohort was male and the median age was 39 (IQR: 34–46). The majority of patients had CD4 counts above 200/µL at transfer (>500/µL 29%, 200–500/µL 55%, <200/µL 15%), and 84% of patients were known to be virologically suppressed.

### Best patient identifier matching

Of 3906 patients, 3498 had one or more records returned by the NHLS. There were a median of 6 records (IQR: 5–7) per patient combining both CD4 and VL data; the maximum was 37 records for one individual. Of these 3498 patients, 3278 (93.7%) were considered confident matches. The distribution of patient identifier match categories is included in table 1. Despite considering multiple potential matching criteria, only 45 additional matches (1.2%; likely and possible matches) were identified using automated procedures. Most of the additional matches (166; 4.7%) were manually confirmed. Only 9 individuals (5.1%) of 175 who required manual review for the best match were not considered a match. Thus, only 0.3% of 3498 with any records were not considered matches. However, an additional 408 (10.4%) of the patients from McCord's HIV clinic did not have records in the NHLS CDW. Thus, overall we were able to match 89.3% of the patients in the McCord record with patients in the NHLS database, and virtually all of the records (99.7%) returned from NHLS were matches to the McCord patients.

### Matching based on CD4 test result and date

After removing the 1 patient who did not have a CD4 test result in the McCord dataset, there were 3451 patients who had ≥1 CD4 records found in the NHLS CDW. Of these 3451 patients, 3270 (94.8%) had CD4 records that

**Table 1** Best match of NHLS data with McCord data solely using patient identifiers

| Matching category (general and specific) | Total=3906, n (%) |
|---|---|
| Confident | 3278 (83.9) |
| Exact match on surname, first name, DOB*, gender | 1823 (46.7) |
| Exact match on surname, at least first word of first name, DOB, gender | 1433 (36.7) |
| Exact match on surname, first name, gender, DOB missing or unusable | 8 (0.2) |
| Exact match on at least first word of surname, at least first word of first name, DOB, gender | 5 (0.1) |
| Exact match on at least first word of surname, at least first word of first name, gender, DOB missing or unusable | 9 (0.2) |
| Likely | 1 (0.03) |
| Surname and first name are reversed, exact match on gender, DOB missing or unusable | 1 (0.03) |
| Likely despite keying errors | 44 (1.1) |
| Exact match on surname, first name, DOB, gender different | 15 (0.4) |
| Exact match on surname, first name, gender, DOB discrepant in one part (day, month or year) | 7 (0.2) |
| Exact match on surname, at least first word of first name, DOB, gender different | 13 (0.3) |
| Exact match on surname, at least first word of first name, gender, DOB discrepant in one part (day, month or year) | 9 (0.2) |
| Possible (manually confirmed 'yes') | 150 (3.8) |
| Exact match on at least first word of surname, first word of first name does not match, exact match on DOB (if usable) and gender (if usable) | 119 (3.0) |
| First word of surname does not match, exact match on at least first word of first name, DOB (if usable) and gender (if usable) | 31 (0.8) |
| Possible (manually confirmed 'no') | 3 (0.08) |
| First word of surname does not match, exact match on at least first word of first name, DOB (if usable) and gender (if usable) | 3 (0.08) |
| Other (manually confirmed 'yes') | 16 (0.4) |
| Other (manually confirmed 'no') | 6 (0.2) |
| No NHLS records | 408 (10.4) |

*DOB, date of birth.

were considered a confident match. Fifty-seven (1.7%) records were considered possible matches and 36 (1.0%) were considered unlikely matches. There were 88 records (2.5%) which did not match on test value and did not match on test date. The distribution of CD4 record matching is shown in table 2.

### Matching based on VL test result and date

After removing 297 patients who did not have VL results in the McCord dataset, there were 244 (6.8%) patients who did not have any VLs found in the NHLS CDW. Among the returned records for the remaining 3365 patients, there were 3306 (98.2%) VL records that were considered a confident match, 11 (0.3%) that were considered possible matches and 1 (0.03%) that was considered an unlikely match. There were 47 records (1.4%) which did not match on test value and did not match on test date. The distribution of VL record matching is shown in table 3.

### Quality of patient identifier match for best test record match

After determining the best match for each test for a specific patient study ID, we assessed how well the patient

identifiers matched on the specific test record. Among the 3469 patients with a confident match on CD4 or VL, 3187 patients (91.9%) were also considered a confident match on the patient identifiers as well, and overall only 10 (0.3%) of these specific test records were not considered matched on the patient identifiers after manual review. For the confidently matched laboratory tests, the possible matches were found to be valid almost all of the time (185/189, 97.9%) after manual review, but only 23/29 (79.3%) of the patient classified other records were valid matches. Most of the additional 272 matches were validated with manual review (208/272, 76.5%). Overall, we manually reviewed 218 records which were confidently matched on a laboratory test, 10 of which were considered not matched (4.6%). The distribution of patient identifier matches by best test matches is shown in table 4.

### DISCUSSION

We assessed the completeness and accuracy of the NHLS CDW by matching patient identifiers and CD4 and VL test

**Table 2** NHLS match for specific CD4 test result and date in the McCord data set

| Matching category (general and specific) | Total=3905, n (%) |
|---|---|
| Confident | 3270 (83.7)* |
| Exact match on CD4 count and test date | 2925 (74.9) |
| Exact match on CD4 count, month and day of test date reversed | 9 (0.2) |
| Exact match on CD4 count, test date within 7 days | 272 (7.0) |
| Exact match on CD4 count, test date discrepant in one part (day, month or year) | 57 (1.5) |
| Exact match on CD4 count and registration date | 3 (0.08) |
| Exact match on CD4 count, registration date within 7 days | 2 (0.05) |
| Exact match on CD4 count, registration date discrepant in one part (day, month or year) | 2 (0.05) |
| Possible | 57 (1.5) |
| Different CD4 counts, exact match on test date | 57 (1.5) |
| Unlikely | 36 (0.9) |
| Exact match on CD4 count, different test date | 36 (0.9) |
| No match | 542 (13.9) |
| Different CD4 counts and different test and registration dates | 88 (2.3) |
| No CD4 value in NHLS | 454 (11.6) |

*Per cents are of the total McCord records with CD4 results.
NHLS, National Health Laboratory Service.

**Table 3** NHLS match for specific viral load test result and date in the McCord data set

| Matching category (general and specific) | Total=3609, n (%) |
|---|---|
| Confident | 3306 (91.6)* |
| Exact match on viral load record and test date | 2993 (82.9) |
| Exact match on viral load record, month and day of test date reversed | 9 (0.2) |
| Exact match on viral load record, test date within 7 days | 254 (7.0) |
| Exact match on viral load record, test date discrepant in one part (day, month or year) | 49 (1.4) |
| Exact match on viral load record, registration date discrepant in one part (day, month or year) | 1 (0.03) |
| Possible | 11 (0.3) |
| Different viral load value, exact match on test date | 11 (0.3) |
| Unlikely | 1 (0.03) |
| Exact match viral load value, different test date | 1 (0.03) |
| No match | 291 (8.1) |
| Different viral load values and different test and registration dates | 47 (1.3) |
| No viral load value in NHLS | 244 (6.8) |

*Per cents are of the total McCord records with viral load results.
NHLS, National Health Laboratory Service.

results from a McCord Hospital dataset to data returned by NHLS for these individuals. NHLS returned records for 89.6% of the individuals requested. Importantly, we found a very low false matching rate in the NHLS data, as only 0.3% of the patients identified by NHLS were not the patients from our initial request. These mismatches may have occurred due to incorrect recording in our internal database, in the NHLS database or incorrect data recorded in the laboratory requisitions. This low false matching rate suggests that our comprehensive matching process is not needed for record reviews for future work. For the few individual patients with mismatching records, there may be implications for missing results when transferring to a new clinic. If there is tight linkage between the NHLS system and public clinic records, these patients may not be correctly found or linked when entering care at a new clinic. Using only personal identifiers, we confidently matched 3278 of 3906 (83.9%) patients. Ignoring identifiers, we confidently matched 83.7% (3270 of 3905) of patients based on CD4 value and test date, and 91.6% (3306 of 3609) of patients with a VL result from McCord Hospital. Of all patients who

had a confident match on either a CD4 or VL test, 91.9% (3187 of 3469) of those specific records were also a confident match using patient identifiers.

Comparing patient identifiers between McCord and NHLS datasets, a vast majority of patients were identified as confident matches. Confident matches made up 94% of the matched cohort, while all other matching categories combined (likely, likely despite keying errors, possible and other) comprised only 6%, suggesting that the overall quality of matched records was high. While it was valuable to examine all potential match types and ranges of match quality, the extensive matching categories may not be necessary as the NHLS records returned were virtually always (99.7%) the patient for whom we requested data. When analysing CD4 and VL test results separately, there was a slightly higher confident matching rate (98.2%) for VL results than for CD4 records (94.8%) among those with any results returned by NHLS. Patients considered a confident match in the CD4 analysis had to have an exact CD4 value match, while patients in the VL analysis had to exhibit a match in VL status if suppressed or exact VL if not suppressed to be considered a confident match. Because VL results for most individuals are grouped into a suppressed category, the CD4 analysis may

**Table 4** Quality of patient identifier match for best test record match

| Patient match category | Record match category (CD4 or viral load)* | | | | |
| | Confident | Possible | Unlikely | No match | Total, n (%) |
|---|---|---|---|---|---|
| Confident | 3187 (91.9%) | 2 (100%) | 9 (100%) | 13 (3.1%) | 3211 (82.2) |
| Likely | 1 (0.03%) | 0 | 0 | 0 | 1 (0.03) |
| Likely despite keying errors | 63 (1.8%) | 0 | 0 | 0 | 63 (1.6) |
| Possible: yes | 185 (5.3%) | 0 | 0 | 4 (0.9%) | 189 (4.8) |
| Possible: no | 4 (0.1%) | 0 | 0 | 0 | 4 (0.1) |
| Other: yes | 23 (0.7%) | 0 | 0 | 0 | 23 (0.6) |
| Other: no | 6 (0.2%) | 0 | 0 | 1 (0.2%) | 7 (0.2) |
| No NHLS records | 0 | 0 | 0 | 408 (95.8%) | 408 (10.4) |
| Total | 3469 | 2 | 9 | 426 | 3906 |

*Percentages are column percentages.
NHLS, National Health Laboratory Service.

provide a more accurate matching process due to the more precise measure of CD4 value.

There are several limitations to our record matching method. NHLS requirements for submitting identifiers with laboratory requisitions during the study period were not strict enough to allow uniformly perfect matching; thus, we had to create extensive matching categories to cover the range of match types and quality. While we considered our patient identifier, CD4 and VL test record matching criteria detailed and comprehensive, a different team might develop an alternative set of rules and designations, and classify specific results differently. Additionally, we had a large range of patient identifier matching criteria for what we considered an adequate match, while these criteria were discussed at length, they ultimately were subjective decisions. While we were able to categorise a large proportion of records by our matching algorithm, there were additional records that we manually reviewed. Although some manual matches could potentially have been more accurately resolved by consulting an outside source, we sought to keep the record matching algorithm self-contained to increase the likelihood that this method could be used by others. Providing laboratory data to NHLS for the matching process might have improved the ability of the NHLS CDW to identify and match our specific patients, so our results might overestimate the ability to match records based solely on patient identifiers. Lastly, while we do not know why 10.6% of individuals requested did not have records returned, we speculate that these individuals may have never had any initial records entered, the data entered may have been so different between NHLS and McCord Hospital that these patients were never identified, or patients may have previously attended a private laboratory.

Despite the drawbacks of this methodology, this study has several important strengths. This is the first analysis to propose a novel method for examining the completeness and accuracy of records related to HIV care from a national data source. We developed a comprehensive and self-contained algorithm that may inform future analyses focusing on linkage to and retention in HIV care. This methodology may also apply to data matching analyses in similar settings, as many sub-Saharan African countries have some sort of national laboratory system.[17] For this matching analysis, we could only include identifiers that were required on the NHLS laboratory requisition form during the study period (first name, surname, gender, DOB). Adding more required identifiers might increase the utility of national laboratory systems for HIV programmes that collect a variety of different identifiers and may also transcend the limitations of using a single official ID, such as South African ID number, for tracking patients across clinics in the public sector. In a previous study where we attempted to collect South African IDs, only a fraction of our participants were able or willing to supply this information and many of the IDs provided were invalid.[18] Lastly, due to the closing of the HIV clinic at McCord Hospital and the rapid transfer of a large cohort of patients, we had a considerable number of comprehensive and up-to-date records with which to assess the quality of NHLS CDW data.

**CONCLUSION**

As South Africa's HIV treatment programme transitions to the public sector and the number of patients initiating ART grows, maintaining a comprehensive and accurate national data repository is of critical importance, as it may serve as a valuable tool to evaluate the effectiveness of the country's HIV care system. Through the method that we created to evaluate national laboratory data, we have demonstrated that the NHLS CDW is both comprehensive and accurate. The NHLS CDW is centralised, broad and supports a wide coverage of public clinics across the country; it, therefore, may serve as an appropriate and effective resource for tracking patients within the public HIV care system. Our ability to confirm the NHLS CDW as a reliable data source can help transcend the limitations of collecting and analysing data within individual clinics, which presents challenges such as differences in

record-keeping methods and marked variability in how patients are identified. Health workers, nurses and clinicians may also be able to use the NHLS to track patients through clinic transfers in the public sector. Additionally, our work suggests that national HIV laboratory systems may benefit from including a more comprehensive set of patient identifiers on laboratory requisition forms to increase the likelihood of containing a complete, accessible list of patients from a wide variety of public HIV programmes. This analysis validates the use of NHLS CDW data in future studies evaluating South Africa's HIV care system and may inform data matching projects in similar settings with national laboratory systems.

**Author affiliations**
[1]Division of Infectious Disease, Massachusetts General Hospital, Boston, Massachusetts, USA
[2]Division of General Internal Medicine, Massachusetts General Hospital, Boston, Massachusetts, USA
[3]Medical Practice Evaluation Center, Department of Medicine, Massachusetts General Hospital, Boston, Massachusetts, USA
[4]Harvard University Center for AIDS Research (CFAR), Boston, Massachusetts, USA
[5]Harvard Medical School, Boston, Massachusetts, USA
[6]Biostatistics Center, Massachusetts General Hospital, Boston, Massachusetts, USA
[7]McCord Hospital, Durban, South Africa
[8]Corporate Data Warehouse, Department of Information Technology, National Health Laboratory Services, Johannesburg, South Africa

**Acknowledgements** We gratefully acknowledge the extensive efforts of the clinical and research teams at Sinikithemba for providing strong leadership during a time of challenging transition.

**Contributors** All authors contributed significantly to this work and reviewed and approved of this manuscript. IVB, principal investigator of this project, led the design and execution of this study as well as all stages of manuscript writing and preparation. MH and RAP led all data analysis efforts. MH initially helped to develop the preliminary novel data cross-matching method, while RAP collaboratively refined the method presented in the manuscript. RAP also contributed substantially to and oversaw all method development. CC and JG both played significant roles in initial data collection and the procurement of records from McCord Hospital. SC also played a significant role in the procurement of CD4 and viral load records from the National Health Laboratory Services, which were used in the data cross-match. SCF, the research assistant, contributed significantly to manuscript writing, editing and review.

**Funding** This work was supported by the National Institutes of Health (R01 MH108427 and R01 MH090326-03S1) and the Harvard University Center for AIDS Research (P30 AI060354), which is supported by the following NIH CoFunding and Participating Institutes and Centers: NIAID, NCI, NICHD, NIDCR, NHLBI, NIDA, NIMH, NIA, NIDDK, NIGMS, NIMHD, FIC and OAR.

**Disclaimer** The content is solely the responsibility of the authors and does not necessarily represent the official views of the National Institutes of Health.

**Competing interests** None declared.

**Patient consent** Not required.

**Ethics approval** The study protocol was approved by the McCord Hospital Research Ethics Committee (Durban, South Africa) and the Partners Human Research Committee (2012-P-001122/1, Boston, Massachusetts, USA).

**Provenance and peer review** Not commissioned; externally peer reviewed.

**Data sharing statement** The data that support the findings of this study are available from the South African National Health Laboratory Services (NHLS) centralised data warehouse (CDW) and McCord Hospital but restrictions apply to the availability of these data, which were used under licence for the current study, and so are not publicly available. Data are however available from the authors on reasonable request and with permission of the NHLS CDW and McCord Hospital.

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
