## [Reviewer comments · BMJ Open]

ARTICLE DETAILS

TITLE (PROVISIONAL)	Assessing the Completeness and Accuracy of South African National Laboratory CD4 and Viral Load Data: A Cross-sectional Study
AUTHORS	Bassett, Ingrid; Huang, Mingshu; Cloete, Christie; Candy, Sue; Giddy, Janet; Frank, Simone; Parker, Robert

VERSION 1 – REVIEW

REVIEWER	Elisabeth Kleppa Oslo University Hospital, Ullevål, Oslo, Norway
REVIEW RETURNED	30-Jan-2018

GENERAL COMMENTS	This is an interesting paper that might be of interest and importance for future studies. It is well written and reads well. I do however have a few comments and suggestions: - Although the aim of the study is clearly understood, I would suggest that the Authors elaborate a little more on the potential use of the results for future work. Why is it important to do this matching?- The method is complex, and as a reader it is somewhat difficult to understand certain aspects of it. Maybe a figure/flowchart with the numbers included/excluded would help?- What did the (few) mismatches imply for the individual patients?- It would be interesting to look into the potential reasons why the mismatches occurred (mistake in the recording in the NHLS data? Missing / incorrect data in the laboratory requisitions?)- Records for 89% of the individuals requested were returned, do we know why there is a considerable fraction missing?
---

REVIEWER	Nei-yuan (Marvin) Hsiao University of Cape Town, South Africa
REVIEW RETURNED	09-Apr-2018

GENERAL COMMENTS	Overall the manuscript is well written and the study has a clear objective and message. I have only a few minor questions around methods which require clarification. 1. The general availability and reliability of patient identifiers in the the setting, in addition to the ones used in the study, were not explained at all in either the background or under methods. Why not use patient folder number, patient national identity number or other potentially unique identifiers for linkage? For example, line 202: " patients with corresponding identifiers (a patient ID internal to our population..." The use of facility identifier I feel need a little description. The basis of this study is facility patient identifier (e.g. hospital folder number), at least alone, is not reliable way of matching/identifying data so background on whether this identifier
--

	was available and its quality is useful for the readers. Do they exist? Are they unique within the facility/province/country? What are they consist of? Were they considered as an identifier in this study? Why were they not considered/not used? 2. Line 205 "The NHLS extracted data in October 2014." Is there a QA process for the CDW data? How was the CDW data selected for matching? Was the McCord patient data matched against the entire CD4 and VL data prior of a particular date range. Is any lab data missing? 3.Data collection and processing and matching paragraphs: This section is very long and often difficult to follow. The convention of inclusion and exclusion criteria is describing the principle applied in methods and state the numbers exclusions in the result section. The numbers are best explained with a (supplementary?) figure. 4. line 218: "We removed 297 patients who did not have VL data in the McCord database, resulting in a cohort of 3609 patients for the VL record matching analysis. " A very brief explanation (pre-ART era?, not linked to care following diagnosis and staging?) of why do these patients have no viral load. This would allow the reader to understand the potential impact of "removed" patients. 5.Line 235 and supplementary table1: "Exact match on at least first word of surname and first names and: DOB (gender missing or unusable) or Gender (DOB missing or unusable)" Has the author reviewed at least some of these confident matches to ensure that they are suitably "confident"? In an hypothetical setting of (male) John N..... (without date of birth) – this often becomes a single variable (first name) matching because gender is often confounded by the first name and first letter of a common letter for surname does not sufficiently narrow the search. In the study this account for very little matching so it does not change the result either way. I feel it should be among one of the lower confidence categories. The study results were fairly discussed and the conclusions were accurate. However, I feel the discussions of other identifiers not used in the study could lend important context as to why this work is important. It could also shed light on how HIV treatment programmes with various identifiers could use their laboratory data as tool of monitor and evaluation.
--	---

VERSION 1 – AUTHOR RESPONSE

Reviewer(s)' Comments to Author:

Reviewer: 1

Reviewer Name: Elisabeth Kleppa

Institution and Country: Oslo University Hospital, Ullevål, Oslo, Norway

Please state any competing interests or state 'None declared': None declared

Please leave your comments for the authors below.

This is an interesting paper that might be of interest and importance for future studies. It is well written and reads well. I do however have a few comments and suggestions:

4.) Although the aim of the study is clearly understood, I would suggest that the Authors elaborate a little more on the potential use of the results for future work. Why is it important to do this matching?

Through our matching study, we have been able to confirm the NHLS CDW as a reliable data source that can help transcend the limitations of collecting data within individual clinics. Health workers, nurses, and clinicians may now be able to confidently use NHLS to track patients through care in the public sector if they do not remain at the same clinic. We have added potential uses of the results of this work to the Conclusion section (lines 449-453):

Health workers, nurses, and clinicians may also be able to use the NHLS to track patients through clinic transfers in the public sector. Additionally, our work suggests that national HIV laboratory systems may benefit from including a more comprehensive set of patient identifiers on laboratory requisition forms to increase the likelihood of containing a complete, accessible list of patients from a wide variety of public HIV programs.

5.) The method is complex, and as a reader it is somewhat difficult to understand certain aspects of it. Maybe a figure/flowchart with the numbers included/excluded would help?

We have included a Supplementary Figure (Figures 1A and 1B) and associated figure legend detailing our process of determining cohorts for the crossmatching analyses and our process of receiving and using NHLS data for the crossmatching analyses.

6.) What did the (few) mismatches imply for the individual patients?

While we are not certain what the few mismatches may imply for individual patients, we speculate that these patients may encounter difficulties, mostly in the form of missing lab results, when transferring to a new clinic. If a public sector clinic relies on NHLS data, these patients may not be correctly found when entering care at a new clinic. In a broader sense, the very low mismatch rate suggests that our comprehensive matching process is not necessary for record reviews for future work. We have included these additional points in the Discussion section (lines 354-358):

This low false matching rate suggests that our comprehensive matching process is not needed for record reviews for future work. For the few individual patients with mismatching records, there may be implications for missing results when transferring to a new clinic. If there is tight linkage between the NHLS system and public clinic records, these patients may not be correctly found or linked when entering care at a new clinic.

7.) It would be interesting to look into the potential reasons why the mismatches occurred (mistake in the recording in the NHLS data? Missing / incorrect data in the laboratory requisitions?)

We do not know why these mismatches occurred, but can speculate that they occurred due to incorrect recording in NHLS or incorrect data entry at the time blood was drawn. We have included this text in the Discussion section (lines 352-354):

These mismatches may have occurred due to incorrect recording in our internal database, in the NHLS database, or incorrect data recorded in the lab requisitions.

8.) Records for 89% of the individuals requested were returned, do we know why there is a considerable fraction missing?

While we do not know why the remaining individuals requested did not have records returned, we speculate that these individuals may have never had any initial records entered, the data entry may have been so different between NHLS and McCord Hospital that these patients were never identified, or patients may have previously attended a private lab. We have included these points in the Discussion section (lines 394-398):

Lastly, while we do not know why 10.6% of individuals requested did not have records returned, we speculate that these individuals may have never had any initial records entered, the data entered may have been so different between NHLS and McCord Hospital that these patients were never identified, or patients may have previously attended a private lab.

Reviewer: 2

Reviewer Name: Nei-yuan (Marvin) Hsiao

Institution and Country: University of Cape Town, South Africa

Please state any competing interests or state 'None declared': None declared

Please leave your comments for the authors below.

Overall the manuscript is well written and the study has a clear objective and message. I have only a few minor questions around methods which require clarification.

9.) The general availability and reliability of patient identifiers in the setting, in addition to the ones used in the study, were not explained at all in either the background or under methods. Why not use patient folder number, patient national identity number or other potentially unique identifiers for linkage? For example, line 202: " patients with corresponding identifiers (a patient ID internal to our population..." The use of facility identifier I feel need a little description. The basis of this study is facility patient identifier (e.g. hospital folder number), at least alone, is not reliable way of matching/identifying data so background on whether this identifier was available and its quality is useful for the readers. Do they exist? Are they unique within the facility/province/country? What are they consist of? Were they considered as an identifier in this study? Why were they not considered/not used?

We have revised the Methods section to provide more clarity about the patient ID, and how our use of this patient ID factored into the patient identifier matching process. The patient ID was a study ID assigned to each patient that was unique to this specific study. We have changed the phrase "patient ID" to "study ID" in the manuscript to help clarify this. The study ID was not a matching criterion, but instead served as a link between the two datasets and was used to keep track of which patients were sent by us to NHLS. When NHLS returned data to us, they included our internal study IDs, which reflected that NHLS returned identifiers that they thought matched with those associated with a specific study ID. We have tried to clarify this process in the "Data Collection and Processing" section of the Methods (lines 199-203):

We sent a list of all 4257 McCord Hospital transfer patients with corresponding patient identifiers (first name, surname, date of birth, gender) to the NHLS for matching of laboratory records (Supplementary Figures 1A and 1B). We also included an internal study ID to identify each patient so that the NHLS could determine which records they were providing matched our requested records.

10.) Line 205 "The NHLS extracted data in October 2014." Is there a QA process for the CDW data? How was the CDW data selected for matching? Was the McCord patient data matched against the entire CD4 and VL data prior of a particular date range. Is any lab data missing?

We have added several sentences in the Methods section to address these questions. We have included information about how the CDW data were selected for matching, the date range over which these data were extracted, and the internal QA process for the CDW data (lines 203-209):

McCord Hospital data were matched against the entire CD4 and VL datasets for KwaZulu-Natal Province from November 1, 2010 through October 31, 2014. To minimize the data lost when exchanging between systems, the NHLS has checks in place to ensure that the number of records sent by the LIS (Laboratory Information System) interface are processed into the CDW. In the event of system failures, there is the ability to re-que data from the LIS. Trend reporting of test volumes over time also assists with data gaps.

11.) Data collection and processing and matching paragraphs: This section is very long and often difficult to follow. The convention of inclusion and exclusion criteria is describing the principle applied in methods and state the numbers exclusions in the result section. The numbers are best explained with a (supplementary?) figure.

We have included a Supplementary Figure (Figures 1A and 1B) and associated figure legend detailing our process of determining cohorts for the crossmatching analyses and our process of receiving and using NHLS data for the crossmatching analyses.

12.) line 218: "We removed 297 patients who did not have VL data in the McCord database, resulting in a cohort of 3609 patients for the VL record matching analysis. " A very brief explanation (pre-ART era?, not linked to care following diagnosis and staging?) of why do these patients have no viral load. This would allow the reader to understand the potential impact of "removed" patients.

We speculate that these patients may not have had a VL lab drawn or may have been recent ART initiators who had not yet been on ART long enough to have met the guidelines for receiving a VL test. We have included this text in the Methods section (lines 222-224):

We removed 297 patients who did not have VL data in the McCord database (missing viral load data may reflect a test not being performed or patients recently initiated on ART who had not yet met guidelines for undergoing a VL test), resulting in a cohort of 3609 patients for the VL record matching analysis.

13.) Line 235 and supplementary table1: "Exact match on at least first word of surname and first names and: DOB (gender missing or unusable) or Gender (DOB missing or unusable)" Has the author reviewed at least some of these confident matches to ensure that they are suitably "confident"? In an hypothetical setting of (male) John N..... (without date of birth) – this often becomes a single variable (first name) matching because gender is often confounded by the first name and first letter of a common letter for surname does not sufficiently narrow the search. In the study this account for very little matching so it does not change the result either way. I feel it should be among one of the lower confidence categories.

We have edited the phrasing of this matching category in both Manuscript Table 1 and Supplementary Table 1 for clarity:

Exact match on at least first word of surname, at least first word of first name: DOB (gender missing or unusable) or gender (DOB missing or unusable).

Using two of the coauthors' names, the following is an example of the type of surname/first name matching that we have deemed applicable to this category. If we had the name "Ingrid Valerie Bassett-Parker", "Ingrid Valerie" would be considered the first name while "Bassett-Parker" would be considered the surname. Based on the category detailed above, "Ingrid Bassett" would be considered a match on at least the first word of the surname, and at least the first word of the first name ("Valerie" is the second word of the first name, while "Parker" is the second word of the surname). In this category, "Ingrid B" would not be sufficient to declare a match.

There are 14 people who fall into this specific category (Table 1), which comprises 0.3% of the entire cohort for patient identifier matching. We have decided to keep these matches in the "confident" category.

14.) The study results were fairly discussed and the conclusions were accurate. However, I feel the discussions of other identifiers not used in the study could lend important context as to why this work is important. It could also shed light on how HIV treatment programmes with various identifiers could use their laboratory data as tool of monitor and evaluation.

We have included additional text in the Discussion section (lines 405-413) discussing the implications of including additional patient identifiers in the study. We were limited by the identifiers we could include in the study, because there were few required identifier fields on the NHLS laboratory requisition form. Adding more required identifiers could increase the utility of national laboratory systems for HIV programs that collect a variety of different identifiers, and may also transcend the limitations of using a single official ID, such as South African ID number, for tracking patients across public sector sites:

For this matching analysis, we could only include identifiers that were required on the NHLS laboratory requisition form during the study period (first name, surname, gender, DOB). Adding more required identifiers might increase the utility of national laboratory systems for HIV programs that collect a variety of different identifiers and may also transcend the limitations of using a single official ID, such as South African ID number, for tracking patients across clinics in the public sector. In a previous study where we attempted to collect South African IDs, only a fraction of our participants were able or willing to supply this information and many of the IDs provided were invalid [19].

FORMATTING AMENDMENTS (if any)

Required amendments will be listed here; please include these changes in your revised version:

1. Supplementary Table Format

- Please re-upload your supplementary files in PDF format.

We have re-uploaded our supplementary files in PDF format.

VERSION 2 – REVIEW

REVIEWER	Nei-yuan (Marvin) Hsiao University of Cape Town South Africa
REVIEW RETURNED	06-Jun-2018
GENERAL COMMENTS	Thank you for promptly addressing the issues raised in the previous review. The methods post-revision is clearer and I am happy with the authors explanations and revision in order to clarify some of the

	questions raised. Overall I think the paper is a valuable contribution to the field of HIV data research and should be accepted for publication.
--	--